# Adaptive Filtering with Fitted Noise Estimate (AFFiNE): Blink Artifact Correction in Simulated and Real P300 Data

**DOI:** 10.3390/bioengineering11070707

**Published:** 2024-07-12

**Authors:** Kevin E. Alexander, Justin R. Estepp, Sherif M. Elbasiouny

**Affiliations:** 1Department of Biomedical, Industrial, and Human Factors Engineering, College of Engineering and Computer Science, Wright State University, Dayton, OH 45435, USA; alexander.201@wright.edu; 2Oak Ridge Institute for Science and Education, Oak Ridge, TN 37831, USA; 3711th Human Performance Wing, Air Force Research Laboratory, Wright-Patterson AFB, Dayton, OH 45433, USA; 4Department of Neuroscience, Cell Biology and Physiology, Boonshoft School of Medicine and College of Science and Mathematics, Wright State University, Dayton, OH 45435, USA

**Keywords:** blink artifact, electroencephalography, adaptive filter, artifact correction, independent component analysis, event-related potential

## Abstract

(1) Background: The electroencephalogram (EEG) is frequently corrupted by ocular artifacts such as saccades and blinks. Methods for correcting these artifacts include independent component analysis (ICA) and recursive-least-squares (RLS) adaptive filtering (-AF). Here, we introduce a new method, AFFiNE, that applies Bayesian adaptive regression spline (BARS) fitting to the adaptive filter’s reference noise input to address the known limitations of both ICA and RLS-AF, and then compare the performance of all three methods. (2) Methods: Artifact-corrected P300 morphologies, topographies, and measurements were compared between the three methods, and to known truth conditions, where possible, using real and simulated blink-corrupted event-related potential (ERP) datasets. (3) Results: In both simulated and real datasets, AFFiNE was successful at removing the blink artifact while preserving the underlying P300 signal in all situations where RLS-AF failed. Compared to ICA, AFFiNE resulted in either a practically or an observably comparable error. (4) Conclusions: AFFiNE is an ocular artifact correction technique that is implementable in online analyses; it can adapt to being non-stationarity and is independent of channel density and recording duration. AFFiNE can be utilized for the removal of blink artifacts in situations where ICA may not be practically or theoretically useful.

## 1. Introduction

Scalp-recorded electroencephalogram (EEG) signals contain contributions from many sources. The exact contributions depend on the relative position and orientation of each source and its recording electrode. Some of the contributing signals are from the neural activity of typical interest with EEGs, but many result from other, undesired physiological and non-physiological “artifact” sources. Many artifact sources can be effectively attenuated with frequency band filters or, in the case of event-related potential (ERP) studies, will be attenuated though epoch averaging if the artifact source signal is not phase-locked across epochs [1]. Signals resulting from eye blinks are one of the most problematic artifacts, however. They occupy the same frequency range of many desired neural signals, so standard filters cannot be used to attenuate them, and because many participants will blink in a nearly time-locked manner to visual stimuli, the artifact often persists even after epoch averaging in ERP analysis. Of particular concern, blink artifacts can distort measurements of the commonly studied P300 ERP, as they share a similar shape, latency, and duration.

Many correction methods have been developed for removing artifacts from EEG data. Early methods often focused on least-squares regression approaches [2] or, in online analyses, using potentiometers to attenuate vertical and horizontal artifact components from the EEG [3], both of which assume that the data observed from any electrode of an EEG is a linear combination of the actual EEG and some scaled contribution from an artifact source [4]. A seminal review article by Croft and Barry [5] is an excellent summary of the history of these regression-based approaches. Following this work, methods that incorporated the concept of dipolar source estimation and reconstruction began to grow in popularity with the recognition that an ocular artifact has a relationship “among all channels”, and not just “between alle channels”, of an EEG, as regression approaches tended to be implemented [6].

One of the most widely used and successful contemporary techniques for removing artifacts from EEG data based on dipolar source estimation and reconstruction is independent component analysis (ICA) [7]. ICA takes a set of EEG signals from multiple recording sites and attempts to separate out the individual sources (i.e., components) based on statistical independence; each estimated component then (linearly) contributes to each recorded electrode. With the EEG deconstructed into (stationary) sources, it may be reconstructed after removing the undesired components such as those containing ocular artifacts. Although widely used and very accurate, there are some theoretical and practical disadvantages to this technique. The primary theoretical disadvantage is the assumption of stationarity in the propagation of sources to the recording electrodes. Stationarity, the assumption that the contributions of source signals to each electrode remain constant over time, is practically, if not effectively, violated as skin conductivity and electrode impedances change over time due to influences from temperature, perspiration, and the drying of the conductive gel or paste between the electrode and skin. The nature of (complex) impedance at the electrode–skin interface also implies that a frequency-dependency may exist in the transfer function between scalp and electrode. However, these theoretical disadvantages are not typically of actual concern in short-duration recordings under carefully controlled laboratory conditions.

The more troublesome issue is the practical disadvantages to using ICA to correct for blink artifacts. For example, ICA is not readily implementable for real-time (i.e., point-by-point) or near-real-time (i.e., online) analyses. ICA is primarily a post hoc analysis, and after it estimates the source space of a recording, the correct component reflecting the blink artifact needs to be selected and removed. This is typically accomplished with review by a subject-matter expert, although there has been great progress toward the automated identification of artifacts (e.g., [8,9]). Further, the ability of ICA to reliably separate sources is highly dependent on the number of data points available to it in both the time and channel dimensions. Short recordings, low sampling rates, and low channel densities will all negatively affect ICA performance. Estimates for the amount of data suggested for ICA are dependent on the channel density and sampling rate [10]; for 64-channel and 256 Hz datasets like those used in this work, the estimate is 320 s. Additionally, the ICA algorithm can estimate no more components than there are recording electrodes; the fewer electrodes there are, the less separation of independent sources into components will be achieved. The problem of electrode density also extends to having too many electrodes, as it is not possible to estimate the number of sources expected from a recording [7] (wherein the number of electrodes would, ideally, be matched). Although in this case, electrodes submitted to ICA can always be reduced in number or dimensionality (e.g., via principal component analysis [10]).

He et al. [11,12] presented an adaptive filter (AF) which continuously (i.e., point-by-point) updates its finite impulse response (FIR) filter coefficients through a recursive least-squares (RLS) algorithm. This “RLS-AF” algorithm was designed specifically to correct simultaneously for vertical (including blinks) and horizontal ocular artifacts with multiple reference noise inputs. To correct for ocular artifacts, RLS-AF takes, at minimum, a single EEG channel signal and a noise reference signal, such as the vertical electrooculogram (vEOG), estimates the correlated noise in the EEG signal, then subtracts that noise estimate from the original EEG signal. Because RLS-AF is updating its filter coefficients point-by-point (i.e., in real-time), it can adapt to any non-stationarities in the source signal’s transfer function. RLS-AF also includes no assumption of frequency-dependency as was demonstrated by He et al. [11]. There are practical advantages of RLS-AF over ICA, the first being that it is not dependent on channel density, as it operates on a single channel at a time, and second, it is easily implemented in real-time. Both of these practical advantages are necessary for brain computer interfaces (BCIs), for example, which often consist of very few recording electrodes which need to be processed as near to real-time as possible.

The primary disadvantage of the RLS-AF method, of its own admission [11], results from bidirectional contamination in the noise reference signal. Just as ocular signals contaminate the EEG, neural signals also contaminate the EOG, which the RLS-AF uses as reference noise input signals. This causes the RLS-AF to erroneously attempt to remove neural signals from the EEG, a phenomenon which is particularly problematic for EEG channels close to the recording EOG electrodes. Removing non-ocular signal sources from the EOG signal would create more ideal reference noise signals. The most obvious methods to achieve this effect are frequency filtering and curve-fitting. Applying a low-pass filter to an EOG signal, however, will only remove noise in the stop band, and any neural activity overlapping in frequency with the artifact will remain in the passband, so bidirectional contamination will still be a problem [11]. Curve fitting is more likely to be able to preserve the underlying blink morphology while smoothing over neural oscillations, thereby reducing cross-talk in the reference noise input that may result in correction errors. Bayesian Adaptive Regression Splines (BARS) [13,14] has previously been suggested as being best suited to this purpose [15]. BARS automatically adjusts the spline knot quantity and location, increasing where amplitude fluctuations are large, to better fit features of interest [15].

Herein, we introduce BARS to the RLS-AF, using a BARS-fitted vEOG as a more ideal reference noise signal. We have named this method AFFiNE (Adaptive Filtering with Fitted Noise Estimation; Figure 1). AFFiNE is presented here as a blink artifact correction method which addresses the theoretical and practical disadvantages of both the ICA and RLS-AF correction methods. AFFiNE makes no stationarity or frequency-dependency assumptions in the artifact propagation to the recording electrodes. The method is not dependent on the number of recording electrodes, is fully automatic, and is implementable as a near-real-time correction technique. In our implementation of AFFiNE, a near-real time was achieved with a maximum delay of 0.75 s (exclusive of computation time) due to the sliding window required for the BARS-fitted vEOG to be calculated. The sliding window required for computation of the BARS-fitted vEOG was 1 s in length (i.e., a 0.25 s overlap); by comparison, the estimated (minimum) window for the computation of the ICA of the same data is 320 s (i.e., a 320 s delay, exclusive of computation time) [10].

The results presented in this work, outlined in a flowchart in Figure 2, demonstrate that the AFFiNE correction method results in improved signal recovery as compared the RLS-AF method and is comparable to that of ICA. We compared three blink artifact correction methods: ICA, RLS-AF, and AFFiNE (constituting a Correction Methods factor) on signals containing two artifact conditions, Blink Present and Blink Absent (constituting an Artifact factor). The three methods are tested in each artifact condition to see if (1) blink artifacts are sufficiently removed when present, and (2) the EEG is sufficiently unaltered when blink artifacts are not present. The methods are examined first in simulated data and then in real EEG data from a traditional visual oddball (i.e., P300) study. As Nunez points out in his chapter in Wolpaw and Wolpaw’s work [16], an algorithm that does not function well under ideal, simulated conditions cannot be expected to improve when subjected to real-world use; in fact, results from the real data analysis did not reflect those from the simulated analysis, and we further discuss the limitations of simulated data and implications for the inclusion of real data from a methodological perspective. Our results show that AFFiNE results in both theoretical and practical improvements in performance over the RLS-AF technique and that its performance is, in practice, indistinguishable from ICA. This filter is an important tool for practical applications of EEGs such as in BCI devices where accurate, online artifact correction is needed and, in particular, when EEG channel density is low.

## 2. Materials and Methods

### 2.1. EEG Recording

#### 2.1.1. Participants

Data from 26 participants (15 males and 11 females; overall mean age of 22.2 years and range of 18–29 years) were evaluated for use in this study. This work was conducted according to the SAGER Guidelines [17]. Although differences by sex in the underlying EEG signals used in this study have been reported (e.g., [18,19]), we do not expect differences by sex in the results of the algorithms used to correct those signals for blink artifacts; accordingly, data and results presented herein were not disaggregated by sex. The sex of all participants was defined based on self-reports and not controlled for in either recruitment or analysis. Given that differences by sex are not expected and sex was not controlled for in this sample, data and results were not disaggregated by sex. Gender was not collected for this study. These data were collected as a part of a larger protocol that was approved by the Institutional Review Boards of Wright State University and the Air Force Research Laboratory; elements of that protocol’s design and additional data collected not relevant to the objectives of this work are not described in detail.

Participants’ data were first evaluated for inclusion into the real dataset to maximize its sample size. The remaining participants were subsequently evaluated for inclusion into the simulated dataset. Of the 26 participants, 17 (11 males and 6 females; overall mean age of 22.1 years and range of 18–28 years) were used in the real-data analysis portion of this study. These participants’ datasets were chosen because they contained a similar number of blink-contaminated and blink-free trials matched among stimulus conditions (this was not true of the remaining 9 participants’ data); these data formed the ‘real dataset’. Of those 9 remaining participants, 8 (3 males and 5 females; overall mean age of 22.5 years and range of 18–29 years) were selected to build a simulated dataset for analyses that cannot otherwise be facilitated using only real data; these data formed the ‘simulated dataset’. The remaining participant’s data was omitted for poor data quality.

#### 2.1.2. Stimuli and P300 Task

To generate the desired P300 ERP, a standard, two-stimuli visual oddball paradigm was used [20,21]. This paradigm presents a participant with sequences of images, being either of a “standard” type that is presented frequently, or of a less-frequently presented “oddball” type. Standard (the capital letter “H”, 1.05° × 1.40°) and oddball (the capital letter “S”, 0.87° × 1.40°) stimuli were shown randomly at a 4:1 ratio; a minimum of two standard stimuli, including the first two stimuli of every sequence, were required between oddball stimuli. A total of 400 stimuli (320 standard, 80 oddball) were presented over a blocked design, in four blocks, as part of the larger experiment. Stimuli were presented to participants on a 24.5″, 240 Hz monitor (BenQ ZOWIE XL2540, BenQ Corporation, Taipei, Taiwan) with a luminance of 5.5 lux at an approximately 32″ viewing distance. Participants were instructed to count “S” stimuli silently and report their count at the end of each block (i.e., participants were not required to make any motor response to stimuli as they were presented). The number of stimuli in each block was always 100, but oddball stimuli counts varied between blocks so that participants could not acclimate to an expected stimulus count. The full stimulus timing sequence is shown in Figure 3.

#### 2.1.3. Recording Equipment

All electrophysiological recordings were made using the BioSemi ActiveTwo system (BioSemi B.V., Amsterdam, The Netherlands). Recordings were made with a 2048 Hz sampling rate at 64 channel locations based on the modified combinatorial nomenclature extension of the 10-10 system [22], excluding the inferior chain except for P9, P10, and Iz [23], with bilateral electrodes on the mastoid process, infraorbital, and outer canthus locations. Task-state and visual stimulus timing information via light sensors placed on the monitor were recorded as events on the ActiveTwo’s 16-bit trigger line (StimTracker 1G, Cedrus Corporation, San Pedro, CA, USA).

### 2.2. EEG Simulation

For an objective measure of correction method performance, the true signal underlying the blink source must be known; since such a ground-truth (herein, shortened to “truth”) is not otherwise attainable from real data, a simulated EEG dataset was created. This simulated dataset, designed to reflect the components in actual, empirical EEG signals, contained a task-irrelevant EEG signal, an ERP signal (i.e., task-relevant), and a blink signal (i.e., artifact-relevant). The creation of each signal is described in detail below; the full, simulated timeseries was generated by the linear summation of these three signals.

#### 2.2.1. Task-Irrelevant EEG: Noise Signal

The BESA Simulator (Version 1.0, June 2013; BESA GmBH, Graefelfing, Germany) “Raw Data Simulation Wizard” was used to create a 615 s task-irrelevant EEG “noise” signal at a 256 Hz sampling rate. BESA assumes a four-shell head model and creates signals with biophysically realistic scalp projections and frequency characteristics using 500 randomly generated, spatially coherent dipoles [24]. In the “Raw Data Simulation Wizard”, shell thicknesses and conductivities were set to default values. Using fully simulated task-irrelevant data ensured that unintended causal linkages to task-relevant or artifact-relevant data were not spuriously generated. However, in order to achieve more accurate morphologies to task-relevant and artifact-relevant data, these two elements were simulated using real data as their basis.

#### 2.2.2. Task-Relevant EEG: ERP Signal

To create the ERP signal, representative P300 samples were needed for both oddball and standard stimulus types, as well as their transfer functions for projection across all simulated electrodes. Single-trial ERPs are ideally desirable for realistic inter-trial variability and morphology; but, as it may be difficult to reliably isolate single-trial ERPs, averaged oddball and standard ERP waveforms were used as the basis for the task-relevant signal instead. An ERP pair (one oddball and one standard trial ERP) was calculated for every participant per block using real data collected from the visual oddball task (see the Stimuli and P300 Task subsection). Before calculating the ERP pairs, the continuous recordings were average-mastoid referenced, bandpass filtered (using guidelines by de Cheveigné & Nelken [25]) to 0.1–30 Hz using a 2nd order IIR Butterworth filter, and corrected for blink artifacts using ICA. Epochs were extracted at Pz from −250 to 800 ms relative to stimulus onset and baseline corrected to the pre-stimulus period. All ERPs were then windowed from 0 to 800 ms so that the P300 peaks of the oddball ERPs were approximately centered and an 800 ms Hanning function was applied to taper the ERP windows to zero at each end. Finally, 30 of these pairs were manually selected based on the visual inspection of pairs with the highest signal quality for both the standard and oddball ERPs. These 30 ERP pairs were from 8 different participants (3 males and 5 females; overall mean age of 22.5 years and range of 18–29 years) and were used as the bases for the task-relevant signal in the simulated data. Example ERP pair waveforms are shown in Figure 4A.

ICA component spatial weights were used to project the simulated ERP signal, calculated at Pz, to other electrode locations. The extended infomax algorithm [26] in the EEGLAB toolbox (v14.1.1) [27] was used to decompose an example recording from one of the participants, after which the P300 component was identified and its spatial weights (i.e., the weights corresponding to that component’s activity from the inverse weight matrix, W^−1^) were used to project the created P300 signal to all simulated electrodes as shown in Figure 4B.

#### 2.2.3. Artifact-Relevant EEG: Blink Signal

The blink signal was created in a manner similar to that of the ERP signal wherein representative examples of blinks from real data were estimated and propagated into the simulated electrodes using a linear transformation based on spatial topography. To obtain blink samples with realistic waveform morphologies and variations, 60 blink samples recorded at electrode FPz (referenced to averaged mastoid) were used. These blink samples were windowed from −250 ms to 250 ms relative to each blink peak. To remove the additional noise and EEG activity in this signal, a smoothing spline with smoothing parameter *p* = 0.99999 was implemented on each sample according to He et al. [12]. Then, a 500 ms Hanning function was applied to each sample to taper the signals to zero at endpoints of the window; an example signal waveform is shown in Figure 4C.

The spatial projection for the blink signal was found using BESA to model two forward facing dipoles at the location of the eyes according to Iwasaki et al. [28]. This spatial projection is shown in Figure 4D and was applied to the blink signal resulting in a realistic projection of the artifact across all simulated scalp electrodes.

#### 2.2.4. Creating Simulated Timeseries Data

The simulated timeseries data were created starting with the 615 s, 69-channel noise signal. The 60 simulated ERP signal samples (30 standard, 30 oddball) were then duplicated (so that the same ERPs could be evaluated for both Blink Present and Blink Absent simulated trials, explained further in the Epoch Sorting by Artifact subsection), and the resulting 120 total ERP samples were then linearly added to the EEG noise signal with 5 s intervals between samples. Each of the 60 blink signal samples were then added such that they coincided with each of the ERP samples. These blink samples were positioned such that they peaked randomly within the 200 to 600 ms window, relative to simulated stimulus onset. Figure 5 shows, at Pz, a 16 s example segment of the resulting simulated signal, containing two oddball and two standard trial samples, one pair of which are contaminated by overlapping blink signals.

### 2.3. Data Analysis

Data analysis was performed using MATLAB (R2017a; The Mathworks, Inc., Natick, MA, USA), utilizing EEGLAB and the ERPLAB plugin (v7.0.0) [29]. All preprocessing and artifact correction methods were identically applied to both simulated and real data.

#### 2.3.1. Pre-Processing

The real datasets were first down-sampled to match the simulated dataset’s 256 Hz sampling rate. Unless noted otherwise, all signals were then referenced to average mastoids, band-pass filtered from 0.1 to 30 Hz using a 2nd order IIR Butterworth filter applied in the forward then reverse directions to achieve a zero-phase filter response [30]. This filter range removes low-frequency drifts and high-frequency artifacts while retaining neural information and preserving the ERP shape [31]. A left eye vEOG was also calculated as the difference between the Fp1 and left infraorbital electrodes.

#### 2.3.2. Artifact Correction Methods

Four correction methods were compared, forming a factor herein referred to as ‘Correction Method’, with four levels: Uncorrected, ICA, RLS-AF, and AFFiNE. The Uncorrected data were only preprocessed according to Section 2.3.1. This section will describe the applications of the remaining three correction methods.

The ICA extended infomax algorithm [26] was the first correction method used after applying only a 1 Hz high-pass 2nd order IIR Butterworth filter in the forward and reverse directions to the raw data (an exception to the pre-processing described in Section 2.3.1). For ICA, all 64 EEG channels, plus the bilateral electrodes on the mastoid process, infraorbital, and outer canthus locations, were included in the definition of the channel space, noting that this channel space included both channels used to define the bipolar vEOG channel used as the reference noise input for both RLS-AF and AFFiNE. The ICA weight matrices obtained from this were then projected onto the 0.1-to-30 Hz band-passed data described in Section 2.3.1. This extra step was taken in an attempt to improve the stationarity of the timeseries data, a best practice for ICA [32] while acknowledging that high-pass cutoff frequencies over 0.3 Hz can result in distorted ERP signals [31]. The ICA component containing the blink signal was manually identified and removed.

The second correction method used was the RLS-AF [11,12]. This filter method was applied using the left eye vEOG signal as a reference noise input. The filter length was chosen as M = 3, and the forgetting factor as λ = 0.999955 (for a 60 s sample window [11]), both of which are consistent with suggested values in the literature [11,12].

The third correction method used was AFFiNE using a BARS-fitted left eye vEOG channel as the reference noise input signal. BARS was fitted to the vEOG signal in one-second sliding windows stepping in 0.75 s increments. The BARS algorithm used was that developed by Wallstrom [15] using the default “Uniform” prior on the number of knots. Also referred to as ‘free-knot splines’ [14], this approach posits that allowing the knots, or endpoints, of piece-wise spline fits to be adjusted in both quantity and location may be preferred in some applications of curve-fitting. In this specific application, where the function of curve-fitting to the vEOG is to preserve (to the extent possible) the morphology of the blink artifact while reducing the contribution of cross-contamination from the EEG in the reference noise input, the knots should be concentrated around the blink artifact [33,34].

#### 2.3.3. Epoch Sorting by Artifact

After artifact correction, all datasets were epoched from −250 ms to 800 ms relative to stimulus onset. Each epoch then needed to be categorized by whether a blink was present or absent in the epoch; this factor, herein referred to as the “Artifact” factor, consists of two levels for “Blink Present” and “Blink Absent”. For the simulated dataset, this was known a priori by design. For the real dataset, categorization was determined from the Uncorrected dataset by manually reviewing all epochs and categorizing each as a “Blink Present” epoch if a blink reached a peak amplitude between 200 ms and 750 ms post-stimulus or as a “Blink Absent” epoch if no part of a blink was observed anywhere in the epoch. If a blink’s peak amplitude occurred outside of this 200-to-750 ms window, or if the epoch contained obvious signals from non-cortical sources, the epoch was discarded from further analysis. Epochs were balanced between Artifact levels to ensure there were equal standard and oddball trial counts in each of the Blink Present and Blink Absent conditions. This resulted in trial counts equal to 60 (30 each of standard and oddball) for the simulated data and a range of 36 to 66 (18 to 33 each of standard and oddball) for participants in the real dataset (Figure 6).

#### 2.3.4. ERP Waveforms and Topographical Errors

For both the simulated and real data, ERP waveforms were plotted for every correction method, as well as both trial types (oddball and standard) and their difference wave (oddball minus standard) at electrodes FPz and Pz. In the simulated data, the truth waveform is also included for comparison with each correction method. However, because the truth waveform is not known in the real data, each waveform can instead be compared to the Uncorrected waveform in the Blink Absent condition as the nearest approximation for the truth waveform.

Mean absolute error (MAE) was also calculated for every channel and plotted as a topographic head plot for each correction method, each trial type (oddball and standard), and their difference wave (oddball minus standard). For the simulated data, MAE was calculated with respect to the known truth data (the signal minus the actual underlying blink signal), averaged across every time point in each of the 30 epochs, then averaged across the 30 epochs. In the real data, MAE was calculated with respect to the Uncorrected data in the Blink Absent condition, averaged across every time point in each participant’s averaged ERP waveform, then averaged across the 17 participants. Note that these result in actual error values in the Blink Absent condition, as any alteration by the correction method would be an error in this condition, but estimated errors in the Blink Present condition since the averaged ERPs in each artifact condition are very similar, but not identical. All topographical maps were plotted using the colormap Viridis, which is a perceptually uniform colormap provided as part of the MatPlotLib toolbox (v2.0.1) [35].

#### 2.3.5. P300 Measurements

For the real datasets, two measures of the P300 response at Pz in both the oddball and difference wave ERPs were taken for comparison across correction methods. These measures were the positive mean amplitude and the 50% positive area latency (referred to as simply amplitude and latency, respectively, from this point onward) and were selected because they reliably quantify the amplitude and latency of the P300 component without being sensitive to high-frequency noise and overlapping negative components [1]. Measurement windows for each waveform, oddball and difference, were used for all participants, across all conditions, to avoid biasing [1]. To choose measurement windows, a collapsed localizer approach [36] was used by examining the grand-averaged ERP across all participants, artifact conditions, and correction methods for each waveform. For the oddball ERP, the measurement window was chosen to be 225 ms to 800 ms. For the difference ERP, the measurement window was defined as 250 ms to 800 ms. ERP measurement errors were calculated for each correction method in each artifact condition using the amplitude and latency measures in the Uncorrected, Blink Absent condition as the truth data. Again, this results in actual error values in the Blink Absent condition, but only estimated errors in the Blink Present condition.

### 2.4. Statistical Analysis

To evaluate the performance of each correction method on the simulated data, errors were calculated at electrodes FPz and Pz for each of the three correction methods and both artifact conditions by subtracting the truth signal from the corrected signal. FPz was chosen because blink artifacts are very large at this location, and Pz was chosen because this is a standard location for P300 measurements. Each correction method was tested in the Blink Present Artifact condition to evaluate how well they corrected blink artifacts in the data and in the Blink Absent Artifact condition to ensure they did not erroneously alter the signals when there was no blink artifact to be removed. Paired *t*-tests were performed on each combination of Correction Method pairs (ICA–RLS-AF, ICA–AFFiNE, and RLS-AF–AFFiNE) within each artifact condition and at both electrode sites, FPz and Pz. 

To evaluate the performance of each correction method on the real data, ERP amplitude and latency measurement absolute errors were calculated relative to the Uncorrected, Blink Absent condition resulting in actual errors in the Blink Absent condition and estimated errors in the Blink Present condition. ERP amplitude and latency measures and their errors were calculated for the 17 oddball and difference waveforms at electrode Pz. Similar to the Simulated data analysis, paired *t*-tests were performed on each combination of Correction Method pairs (ICA–RLS-AF, ICA–AFFiNE, and RLS-AF–AFFiNE) within each artifact condition. All statistical tests were performed using IBM SPSS Statistics 24 using α = 0.05 with no correction made for multiple comparisons.

## 3. Results

### 3.1. Simulated Data Results

#### 3.1.1. ERP Waveforms

To visually observe the simulated P300 and blink artifact, and the effects of all Correction Methods on both event-related waveforms (standard, oddball, and difference wave) time-locked to stimulus onset were averaged and plotted as timeseries at representative electrode locations. The oddball, standard, and difference wave ERPs, calculated from all Artifact by Correction Method combinations using the simulated dataset, are shown at electrodes FPz and Pz in Figure 7. ERPs calculated from only the simulated cortical source (i.e., task-relevant and -irrelevant) data, labeled as ‘Truth’, are shown for comparison. The electrodes FPz and Pz were chosen as these are complementary locations with respect to blink and P300 amplitudes. Pz is the quintessential electrode location for observing the P300; however, blinks tend to be smaller in amplitude at Pz because it is (relatively) distant from the eyes (as compared to other electrodes). Conversely, blink amplitudes are quite large at FPz, but the P300 is negligibly observable. Qualitatively, high-amplitude blink artifacts are clearly seen in the stimulus-locked ERPs, including the difference wave. The appearance of blink artifacts can be clearly seen in the oddball and standard ERPs, as well; however, the appearance of the blink artifact in the difference wave ERP at Pz is less noticeable. From this visual inspection of the simulated data results, the following can be stated: (1) the simulated data realistically represented both P300 and blink artifact morphologies, (2) all Correction Methods appear to generally function to remove the blink artifact, and (3) even when a blink artifact is not present, the resulting methods can result in changes in the underlying data.

#### 3.1.2. Topographical Errors

To visualize any spatial differences between the Correction Methods in both the Blink Present and Blink Absent conditions, MAE error values for all Artifact by Correction Method combinations were plotted for the 64-channel electrode array in a two-dimensional projection of the scalp surface (Figure 8). Qualitatively, RLS-AF results in noticeably higher error amplitudes, particularly for frontal electrodes (i.e., those electrodes closest to the origin of the blink artifact), whereas both ICA and AFFINE produce more spatially consistent errors throughout the topography. Although less pronounced, it does appear that AFFiNE results in higher errors than ICA, particularly in the Blink Present condition. Taken collectively across all Correction Methods and Artifact Conditions, there are sufficient qualitative differences in MAE to justify further quantitative analyses.

#### 3.1.3. Analysis at FPz and Pz

To quantitatively compare all Correction Methods and Artifact Conditions in the simulated data, MAE values for the difference wave data shown in Figure 8 were plotted as ‘swarmplot’ distributions and compared using paired *t*-tests. The difference wave data were chosen as the difference wave is typically of primary interest for analysis in most P300 studies; in addition, the difference wave also evidenced the strongest spatial effect on correction error based on the visual inspection of the results in Figure 8. Where Figure 8 showed the MAE distribution averages, the full MAE distributions for electrodes FPz and Pz are shown in Figure 9 for each Artifact Condition and Correction Method. Each data point is the MAE value for each of the 60 trials. Interestingly, where AFFiNE errors at FPz were improved on average from the Blink Present to the Blink Absent conditions (2.605 μV to 1.581 μV), the RLS-AF errors changed very little (4.473 μV to 4.228 μV), indicating that the RLS-AF continued to alter the data even when there was no blink artifact to be corrected. The *t*-test results are shown in Table 1. In every comparison, AFFiNE achieved statistically significantly smaller errors than RLS-AF, and ICA errors were statistically significantly smaller than both RLS-AF and AFFiNE. For simulated data, these results demonstrate an improvement on RLS-AF using AFFiNE, as hypothesized, but AFFiNE was not able to produce errors in the artifact-corrected data as low as those of the ICA.

### 3.2. Real Data Results

To identify any assumptions inherent in the simulated data that may affect results in a manner that is not consistent with what would be expected in real-world situations, we extended our assessment and comparison of the three artifact correction methods to real data. Practically speaking, real data results should be of higher interest to practitioners since any artifact correction algorithm is intended to be used on real data. However, the interpretation of results in real data is often more difficult because it is not possible to observe the underlying EEG unaltered by artifacts in all but the most extreme of situations (e.g., [37]). Although direct comparisons of altered versus unaltered EEG data are impossible in real data, comparisons between data that do and do not appear to be contaminated by artifacts (determined by visual inspection) can be made; this is the approach presented herein for the analysis of the real data from the visual oddball task. All analyses performed on the simulated data have similar analyses for the real data.

#### 3.2.1. ERP Waveforms

As with the simulated data, to visually observe the real P300 and blink artifact, and the effects of all Correction Methods on both, event-related waveforms (standard, oddball, and difference wave) time-locked to stimulus onset were averaged and plotted as timeseries at representative electrode locations. The grand-averaged ERPs across all 17 participants calculated from the real datasets are shown at electrodes FPz and Pz in Figure 10. Both P300 and blink artifact morphologies are similar to what was observed in the simulated data. Qualitatively, high-amplitude blink artifacts are clearly seen in the stimulus-locked ERPs, including the difference wave; however, the appearance of the blink artifact in the difference wave ERP at both FPz and Pz is more pronounced than in the simulated data. From this visual inspection of the simulated data results, the following can be stated: (1) the real data generated both P300 and blink artifact morphologies as expected and designed, (2) all Correction Methods appear to generally function to remove the blink artifact, and (3) even when a blink artifact is not present, the resulting methods can result in changes in the underlying data. Also worth noting is the similarity of the Uncorrected, Blink Absent data waveform morphologies to those of the Corrected, Blink Present waveforms, which strongly suggests that the choice selection of the trials to be included in the Uncorrected, Blink Absent condition was adequate to perform the desired comparative analyses (i.e., lacking a truth signal for artifact-free waveforms in real data).

#### 3.2.2. Topographical Errors

As with the simulated data, to visualize any spatial differences between the Correction Methods in both the Blink Present and Blink Absent conditions, error values in the real data were calculated using the Uncorrected data in the Blink Absent condition as the “truth data” and plotted for the 64-channel electrode array in a two-dimensional projection of the scalp surface (Figure 11). These errors were calculated as MAE values for each participant’s individual ERP for the oddball, standard, and difference waves. In contrast to the simulated data, there are no obvious spatially correlated distortions in any Correction Method or Artifact Condition. Overall, errors appear to be higher in the Blink Present condition than in the Blink Absent condition, but still practically small (low, single-digit µV) in amplitude. Taken together with the same analyses for the simulated data, these results highlight the differences between real data and simulated data that motivate the comparative analysis between them.

#### 3.2.3. Measurement Errors in the Blink Absent Condition

Following the quantitative analysis performed on the simulated data, it reasons that a similar analysis for comparative purposes is logical for the real data as well. To quantitatively compare all Correction Methods and Artifact Conditions in the real data, MAE values for the difference wave data shown in Figure 11 were plotted as ‘swarmplot’ distributions (Figure 12) and compared using paired *t*-tests. In addition to the difference wave, results from the oddball response were also quantitatively compared in the event that the difference wave calculation produced atypical results due to any effects that may have resulted from the subtraction of the standard from the oddball response (i.e., because the properties of the data could not be as carefully controlled as in simulated data). Analyses for the Blink Absent and Blink Present conditions are discussed separately as the truth signal to which each were compared were different (necessitated by the analysis of real, as compared to simulated, data).

In the Blink Absent condition, actual errors were calculated relative to the Uncorrected data. Figure 12 shows the amplitude (Figure 12A) and latency (Figure 12B) measurement absolute error distributions for all 17 participants for both the oddball and difference waves. To compare the measurement errors resulting from each Correction Method, paired *t*-tests were conducted between each Correction Method for both the oddball and difference wave amplitude measures. Latency error distributions were bimodal and not appropriate for this statistical test; practically speaking, at a sampling rate of 256 Hz (i.e., 3.9 ms between sampling points), the errors in latency measurement were either zero or one sample point, with rare exceptions. The results of the paired *t*-tests are shown in Table 2, which indicate that, for both the oddball and difference waveform, AFFiNE achieved smaller amplitude measurement errors than the RLS-AF correction method. There was no significant difference from ICA in any comparison. These results illustrate that, as hypothesized, AFFiNE offers a statistically significant reduction in correction error (for P300 amplitude) in the Blink Absent condition as compared to RLS while simultaneously achieving a correction error not statistically significantly different from ICA, which is the gold standard, in real data.

#### 3.2.4. Measurement Errors in the Blink Present Condition

In the Blink Present condition, measurement errors were estimated using the measurements in the Uncorrected, Blink Absent condition as an estimate of the “truth values”. Figure 13 shows the 17 amplitude (Figure 13A) and latency (Figure 13B) measurement errors arising from the oddball and difference waveforms in the Blink Present condition. Table 3 shows the results of paired *t*-tests comparing the measurement errors of each correction method. For both the amplitude and latency measures, no correction method resulted in significantly different measurement errors in the Blink Present condition. Collectively, these results show that all three Correction Methods performed equally (for both amplitude and latency) at removing blink artifacts in the Blink Present condition.

For completeness, individual participant ERP metrics for amplitude and latency, which inform the data in Figure 12 and Figure 13, are shown in Table 4, Table 5, Table 6 and Table 7.

## 4. Discussion

In this paper, we presented a comparison of three Correction Methods for removing blink artifacts from EEG data in both simulated and real datasets. Our specific goal was to present and evaluate an improvement on the RLS-AF method, AFFiNE (Figure 1), which seeks to improve upon the cross-talk problem in the reference noise input signal. We compared both RLS-AF and AFFiNE against ICA primarily to evaluate their blink artifact removal performance against a known and accepted gold standard, but also to demonstrate situations in which RLS-AF or AFFiNE, depending on their performance, may be a preferred solution for both theoretical and practical reasons. As a practical example of a highly utilized EEG signal that is often corrupted by blink artifacts, we chose the P300 (or, more specifically, the P3b; Figure 4) ERP. In simulated data, our comparison of Correction Method performance was straight-forward and based on amplitude error in the corrected time-domain signals from a known “truth” signal. Without the availability of a “truth” signal in real data, we evaluated the Correction Methods against P300 epochs in which no blink artifacts were observed. Specifically, P300 amplitude (positive mean amplitude) and latency (50% positive area latency) measures were compared between (balanced) epochs that were corrected for blink artifacts and epochs that did not contain blink artifacts to begin with.

### 4.1. Simulated Data Results

The three Correction Methods, ICA, AFFiNE, and RLS-AF, were tested first in simulated EEG data. Simulated EEG data are advantageous because amplitude errors from “truth” values of the time-series (i.e., voltage) data can be obtained and the performance of each Correction Method can be objectively compared. In the simulated data tests, each of the three Correction Methods were applied to a signal containing blink artifacts. MAEs for each of the three Correction Methods were calculated for 60 epochs that had contained blink artifacts (Blink Present) and 60 epochs that were free of blink artifacts (Blink Absent). As can be seen in Figure 9 and Table 1, the AFFiNE filter method resulted in statistically significantly lower errors than the RLS-AF Correction Method in the Blink Present condition, which reflects the problem of cross-talk in the vEOG signal and the effectiveness of AFFiNE at correcting for this problem. This cross-talk problem is further demonstrated by the statistically significantly higher signal errors in RLS-AF as compared to AFFiNE in the Blink Absent Artifact condition, also in Figure 9 and Table 1. This indicates that, even when there is no ocular artifact present in the data (known by design in the simulated data), there are still signals of other sources (including neural) present in the vEOG noise reference signal that RLS-AF attempts to remove from the EEG. Although the error resulting from ICA as compared to AFFiNE was statistically significantly lower in both the Blink Present and Blink Absent conditions, AFFiNE performed to within 1 µV of ICA, on average. It should be noted that ICA could have had an advantage in these analyses because the simulated data were stationary by design, which is an assumption of ICA (but not RLS-AF or AFFiNE), and this assumption is not necessarily met in real data, but is often satisfied in preprocessing steps prior to the application of the ICA algorithm itself [10].

### 4.2. Real Data Results

These same three Correction Methods were again compared in real recorded EEG data, but as the desired “truth” signal cannot be wholly known in real data, a different approach for Correction Method comparison was needed. Thus, half of the real data epochs (Figure 6) were selected to contain blink artifacts occurring within the P300 time window (Blink Present) and the other half were selected to contain no blink artifacts (Blink Absent). Analyzing the P300 ERP showed the effect of blinks at Pz (Figure 10) and demonstrated why it is important to correct for them. Each of the three Correction Methods resulted in measures of amplitude and latency similar to those of the artifact-free data prior to filtering (Figure 12 and Figure 13). Statistical analyses (Table 2 and Table 3) showed that all three correction methods resulted in distributions of amplitude and latency metrics that were not significantly different from any other method, either within or between artifact conditions, with one exception: when blinks were absent, AFFiNE resulted in statistically significantly lower error than RLS-AF in the measurement of blink amplitude. This is the same effect of cross-talk observed in the simulated data, which becomes more prominent as an area of concern with RLS-AF when blinks are absent from the analyzed epochs.

### 4.3. Algorithm Performance Comparison in Simulated vs. Real Data

The difference between the results in the simulated and real data highlight an important concern in evaluating artifact correction methods: although data conditions are more ideally known in simulated data, and even when utmost care is taken when considering the assumptions and properties of the simulated data, it may be possible that real data do not produce similar results. In our simulated data analyses, ICA was consistently statistically significantly better at artifact correction than both RLS-AF and AFFiNE in all comparisons; however, in real data, the only statistically significant improvement was found in AFFiNE as compared to RLS-AF in the Blink Absent condition. This suggests that, when possible, it is prudent to carefully design comparative analyses in real data conditions to most meaningfully evaluate artifact correction differences between methods. Results based on simulated data alone should be heavily caveated with this in mind. It should be noted that the visual oddball paradigm used for real data collection in this study was not strictly designed with this analysis in mind; however, through careful thought and data inspection, a suitable case for objectively comparing artifact correction performance in real data was designed post hoc. This is the likely outcome in similar situations given that it is unlikely that the presence of an artifact and the EEG response of interest can be designed to be elicited independently of each other a priori, but in essence, this is the process that must be undertaken. For external artifacts, e.g., motion or environmental artifacts, an a priori independent design may be feasible; for internal artifacts, e.g., artifacts from other physiological sources, this is less likely to be achievable.

### 4.4. Differences between Simulated and Real Data

It is interesting to note that, for the difference wave ERP in Figure 10, the presence of blink artifacts is less noticeable as compared to the individual oddball and standard responses. This result, however, is not entirely surprising, given that the number of trials for both the Blink Present and Blink Absent conditions were balanced for all participants (Figure 6). This balancing was done specifically to remove the effects of bias due to uneven occurrence rates of stimuli (i.e., 20% oddball versus 80% standard presentation) and the potential that, for this particular dataset, the occurrence rates of blink artifacts may differ between stimulus types (although we did not investigate the presence or absence of this effect specifically, we simply controlled for it regardless). The likely effect of balancing this for the purposes of artifact removal analyses is that each participant experienced approximately the same distribution of blinks, in both amplitude and phase, relative to stimulus onset, such that the difference wave nearly eliminated their average effect on both the oddball and standard ERPs. This interpretation is reasonably supported by the individual participant ERP waveforms from Pz in Figure 14 and Figure 15, where noticeable effects in the Blink Present condition (as compared to the Blink Absent condition) are observed in both the standard and oddball ERP waveforms but substantially mitigated in the difference wave ERP. The same effects do not appear to be consistent at Fpz, where the amplitude of the blink artifact is much higher, thus indicating that this artifact balancing effect observed in Pz is at least in part due to the comparatively weak effect of blink artifacts experienced at that electrode location. It would not be expected for these results to be consistent in all difference wave analyses, but they are logical here, where we carefully chose our data to study the effects of the artifact correction approaches, and not the oddball paradigm itself. Although the design of our simulated dataset was well-motivated for this purpose, from the simulated data alone, it would be possible to come to the incorrect conclusion that blink artifacts are sufficiently mitigated in difference wave analyses; we highlight this point to further illustrate that, despite their benefits, simulated data often have unintentional limitations that can be elucidated when compared to real data. Where possible, the use of both simulated and real data in the development and evaluation of signal processing techniques is a recommended best practice.

### 4.5. Extension of AFFiNE to Other Artifact Types

Although only blink artifacts were evaluated in this study, AFFiNE is theoretically extensible to eye movement artifacts as well. Eye movement artifacts were not readily present in the P300 data chosen to evaluate blink artifact removal, but from a practical perspective, the extension for evaluation in both synthetic and real data and the inclusion of additional reference noise input signals would follow this work and the work of He et al. [11,12]. From a similar perspective, it is also possible that AFFiNE would be suitable for the removal of other artifact sources, including but not limited to head motion, body motion, and low-frequency skin potentials. In short, AFFiNE should work well for artifact removal when those artifact sources can be represented by an independent reference noise input signal that uniquely characterizes the spatiotemporal dynamics of the artifact. We expect that some artifact sources, like muscles, are not solvable with AFFiNE, as the spatial source(s) of muscle artifacts are broad and temporally overlapping in many situations.

## 5. Conclusions

Blink artifacts are one of the most common artifacts in EEGs and are problematic in ERP studies. ICA is one of the most common “gold standard” methods for correcting this artifact; however, this method can be impractical, as it does not typically perform well with short recordings or low channel densities and is not well-suited for online implementation. Reference-based adaptive filters like the RLS-AF are often more practical in these situations as it is intended for online implementation; it is a point-by-point (i.e., real-time) process; and it is not dependent on channel density. Being an adaptive filter that updates its FIR filter coefficients with the same point-by-point process, it has the advantage of not assuming stationarity, which is a potential limitation of the ICA technique. The RLS-AF correction method uses a bipolar vEOG signal as a reference noise input, however, which results in the problem of cross-talk, where some amount of EEG is inherently present and effects on the EEG signal where there should be none are therefore observed. Herein, AFFiNE was created and introduced to mitigate this cross-talk problem by resolving a more ideal reference noise input signal (i.e., an EEG free of residuals that are not desirable for its purpose). As we have showed, the AFFiNE filter results in statistically significantly lower errors as compared to the RLS-AF technique in both simulated (Section 4.1) and real (Section 4.2) data, with the most important result being that AFFiNE reduces the effect on the EEG response where there should be none (i.e., when a blink artifact is not present; Section 3.2.3). AFFiNE results in errors either practically or observably comparable to ICA (Section 3.2.3 and Section 3.2.4) with the advantage of computational speed, independence with respect to EEG channel density, and freedom from the assumption of stationarity, as discussed in the Introduction. AFFiNE can be implemented in near real-time with a processing delay (approximately 0.75 s in this study) that would be negligible for many applications. Additionally, our results highlight the need to carefully caveat artifact correction algorithm performance evaluated only on simulated data and suggest that careful considerations of methods that allow for objective comparison of algorithm performance in real data are needed, in addition (Section 4.3 and Section 4.4). In sum, AFFiNE is a practical alternative to ICA for online data analysis and a theoretical alternative to ICA in situations where data stationarity may be a concern.

## Figures and Tables

**Figure 1 bioengineering-11-00707-f001:**
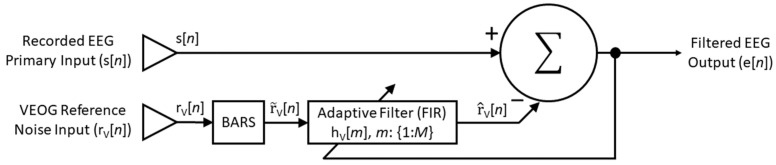
Block diagram of the AFFiNE method. The reference noise input, r_v_[*n*], is conditioned by BARS to be free of bidirectional contamination from EEG sources close to the recording VEOG electrodes, thus resulting in a more ideal reference noise input signal for RLS-AF. The resulting adaptive filter is a FIR of length M, which operates on the conditioned reference noise input signal and is then linearly subtracted from the recorded EEG signal, s[*n*], to produce an EEG signal that is free of blink artifacts, e[*n*].

**Figure 2 bioengineering-11-00707-f002:**
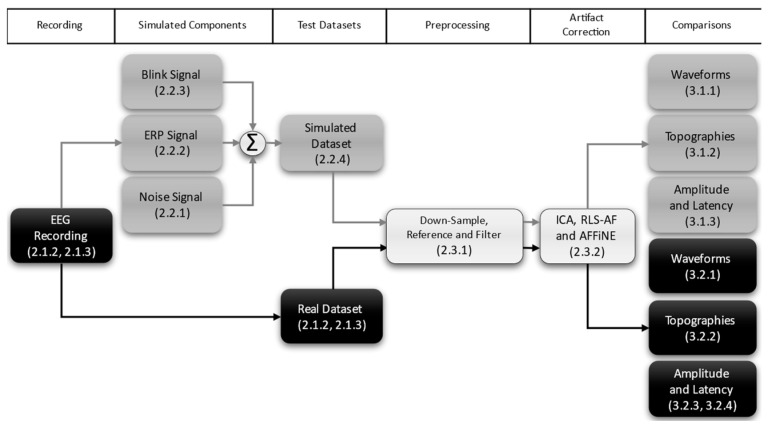
This flowchart graphically outlines the main components of the current study with the relevant section numbers provided in parenthesis. Those sections primarily pertaining to the simulated data are shown in grey, and those pertaining primarily to the real data are in black; sections common to both datasets are shown in white.

**Figure 3 bioengineering-11-00707-f003:**
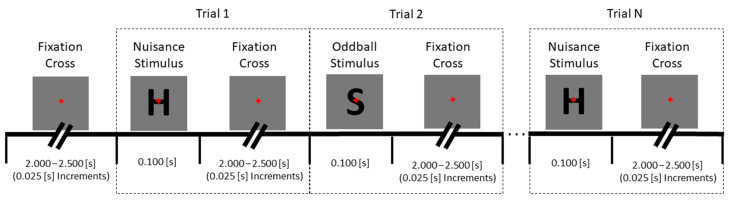
Stimulus timing sequence for the visual oddball paradigm. Each stimulus appeared on the screen for 100 ms, with inter-stimulus intervals varying on a uniform distribution between 2.0 and 2.5 s. During the inter-stimulus interval, a fixation cross appeared on the screen.

**Figure 4 bioengineering-11-00707-f004:**
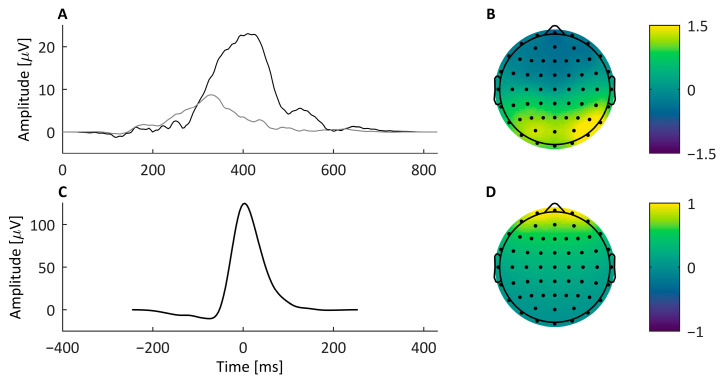
Example signals used in the creation of the simulated dataset. (**A**) Example standard and oddball ERPs (grey and black, respectively) at electrode Pz, and (**B**) the spatial weights used to project the signal to all simulated electrodes. (**C**) An example simulated blink signal at FPz, and (**D**) its spatial weights used to project the signal across the scalp.

**Figure 5 bioengineering-11-00707-f005:**
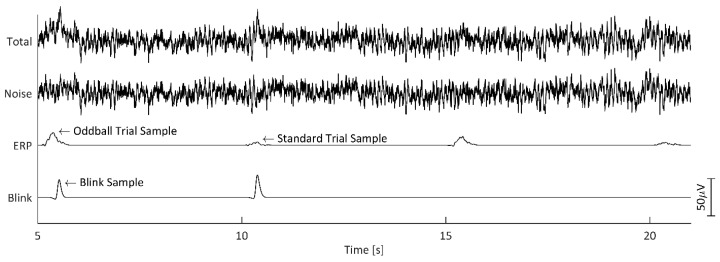
Example time segment of the simulated signals and the summed (‘Total’) simulated signal at electrode Pz. The total simulated signal consisted of summed ‘Noise’, ‘ERP’, and ‘Blink’ signals. The blink samples in the blink signal were placed such that they coincided with half of the ERP samples.

**Figure 6 bioengineering-11-00707-f006:**
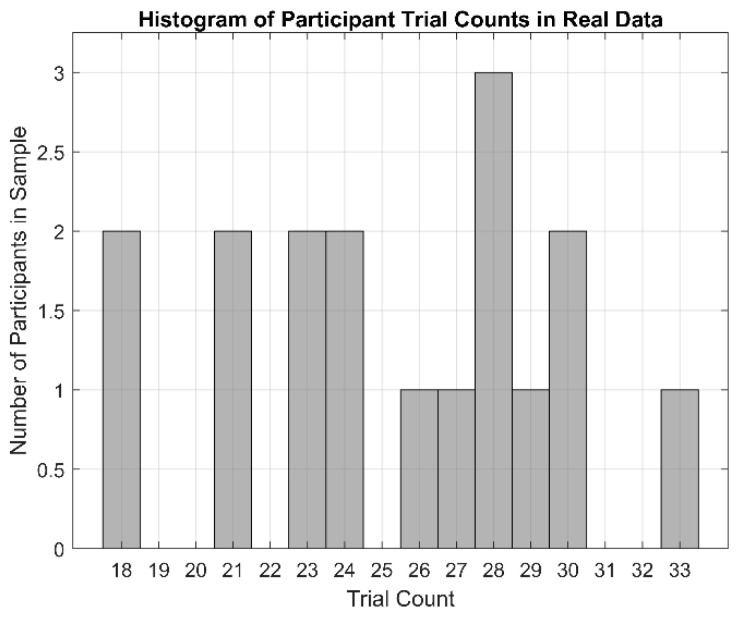
Histogram showing trial counts for each participant contained in the sample of real data used to compare the performance of the correction methods. A total of 17 participants were included with trial counts ranging from 18 (2 participants) to 33 (1 participant).

**Figure 7 bioengineering-11-00707-f007:**
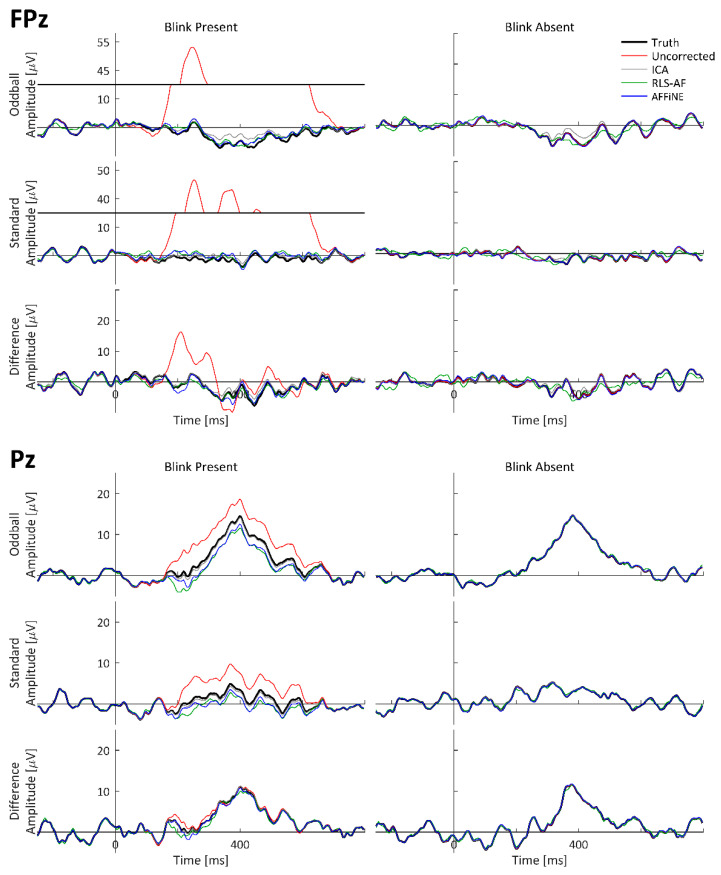
Standard, oddball, and difference wave ERPs calculated from the simulated dataset for all Artifact by Correction Method combinations at FPz and Pz. Because these are simulated data, ‘Truth’ is the actual ERP calculated in each condition using only the simulated cortical source (i.e., task-relevant and -irrelevant) data. Data are shown with y-axis breaks in FPz/Blink Present to visualize the Uncorrected blink artifact maximum amplitude without detracting from the lower-amplitude features of the Truth and Corrected (ICA, RLS-AF, and AFFiNE) waveforms.

**Figure 8 bioengineering-11-00707-f008:**
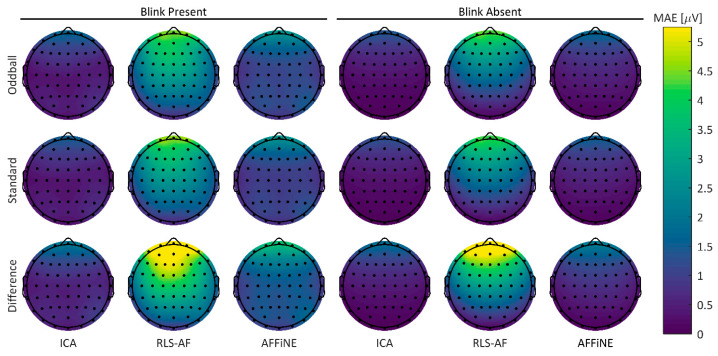
Topographic MAE from each of the standard, oddball, and difference wave ERPs, both Blink Present and Blink Absent Artifact conditions, and the ICA, RLS, and AFFiNE levels of the Correction Method. The ‘Uncorrected’ level of the Correction Method is omitted here because no correction is reasonably attempted in the Unfiltered condition.

**Figure 9 bioengineering-11-00707-f009:**
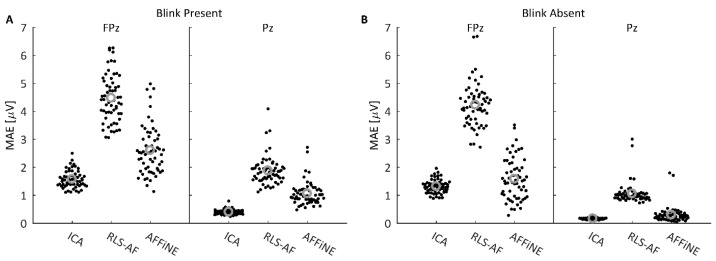
Simulated data MAE values at FPz and Pz electrodes over 30 oddball and 30 standard trials in the Blink Present (**A**) and Blink Absent (**B**) Artifact conditions. Individual MAE values are plotted as a ‘swarmplot’ to aid in the visualization of the discrete data by offsetting data points in the x-dimension; this offset in the x-dimension is for visualization only and does not encode any information about the distribution of the presented data. Grey circles indicate distribution means.

**Figure 10 bioengineering-11-00707-f010:**
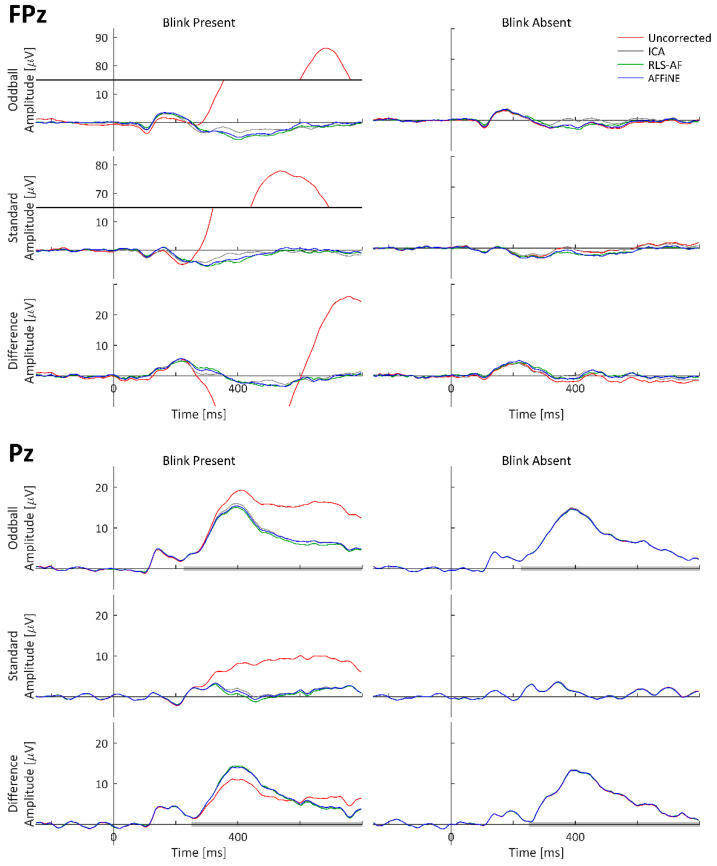
Real data, original and corrected, grand-averaged ERP waveforms at electrodes FPz and Pz. Pz Oddball and difference wave ERP measurement windows are indicated as shaded portions of the x-axis. Data are shown with y-axis breaks in FPz/Blink Present to visualize the Uncorrected blink artifact maximum amplitude without detracting from the lower-amplitude features of the Truth and Corrected (ICA, RLS-AF, and AFFiNE) waveforms.

**Figure 11 bioengineering-11-00707-f011:**
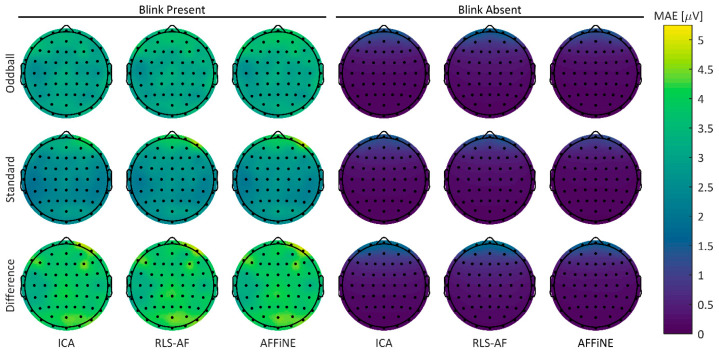
Real data mean participant MAEs. Errors were calculated using the Uncorrected data in the Blink Absent Artifact condition as the “truth data”.

**Figure 12 bioengineering-11-00707-f012:**
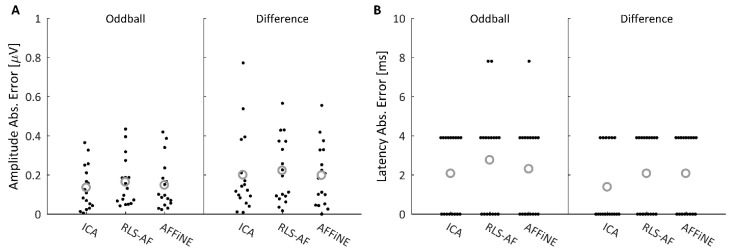
Oddball and difference wave ERP amplitude (**A**) and latency (**B**) measure absolute errors after performing each correction method on the Blink Absent trials. Individual absolute errors are plotted as a ‘swarmplot’ to aid in the visualization of the discrete data by offsetting data points in the x-dimension; this offset in the x-dimension is for visualization only and does not encode any information about the distribution of the presented data. Grey circles indicate distribution means. Practically speaking, latency error was either zero or one sample point (i.e., 3.9 ms), with rare exceptions, across all Correction Methods.

**Figure 13 bioengineering-11-00707-f013:**
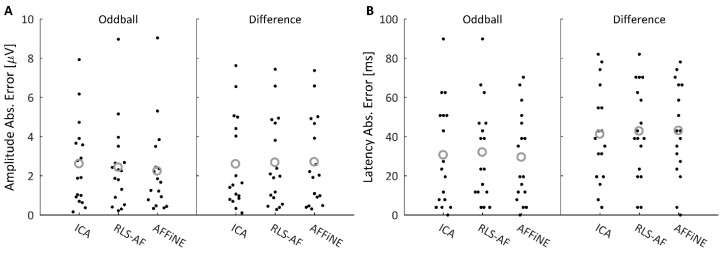
Oddball and Difference wave ERP amplitude (**A**) and latency (**B**) measure absolute errors after performing each correction method on the Blink Present trials. Individual absolute errors are plotted as a ‘swarmplot’ to aid in the visualization of the discrete data by offsetting data points in the x-dimension; this offset in the x-dimension is for visualization only and does not encode any information about the distribution of the presented data. Grey circles indicate distribution means.

**Figure 14 bioengineering-11-00707-f014:**
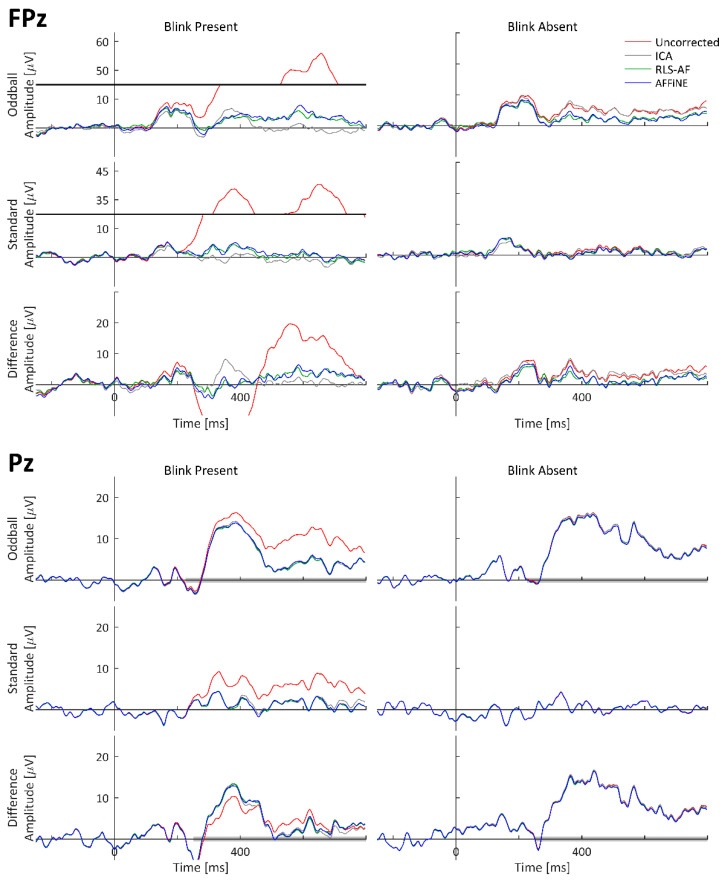
Real data from a single participant; uncorrected and corrected ERP waveforms at electrodes FPz and Pz. Pz Oddball and difference wave ERP measurement windows are indicated as shaded portions of the x-axis. Data are shown with y-axis breaks in FPz/Blink Present to visualize Uncorrected blink artifact maximum amplitudes without detracting from the lower-amplitude features of the Truth and Corrected (ICA, RLS-AF, and AFFiNE) waveforms.

**Figure 15 bioengineering-11-00707-f015:**
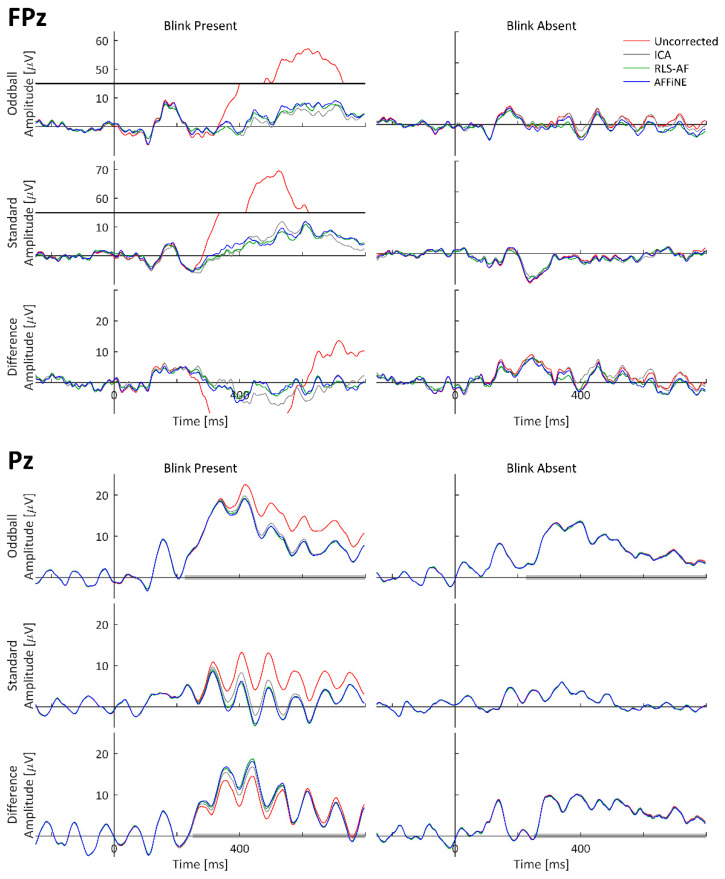
Real data from an additional, single participant; uncorrected and corrected ERP waveforms at electrodes FPz and Pz. Pz Oddball and difference wave ERP measurement windows are indicated as shaded portions of the x-axis. Data are shown with y-axis breaks in FPz/Blink Present to visualize Uncorrected blink artifact maximum amplitudes without detracting from the lower-amplitude features of the Truth and Corrected (ICA, RLS-AF, and AFFiNE) waveforms.

**Table 1 bioengineering-11-00707-t001:** Paired *t*-test results for correction method MAE values in each Artifact Condition at electrodes FPz and Pz. Asterisks indicate significant differences at the 0.05 level. In every case, ICA achieved the smallest errors, followed by AFFiNE, with the RLS-AF resulting in the highest errors.

Artifact Condition	Electrode	Test Pairs	M	SD	Statistic	*p*	
Blink Present	FPz	ICA–RLS-AF	−2.901 μV	0.887 μV	*t*(59) = −25.323	<0.001	*
		ICA–AFFiNE	−1.033 μV	0.956 μV	*t*(59) = −8.372	<0.001	*
		RLS-AF–AFFiNE	1.868 μV	0.669 μV	*t*(59) = 21.621	<0.001	*
	Pz	ICA–RLS-AF	−1.479 μV	0.491 μV	*t*(59) = −23.353	<0.001	*
		ICA–AFFiNE	−0.658 μV	0.383 μV	*t*(59) = −13.329	<0.001	*
		RLS-AF–AFFiNE	0.821 μV	0.247 μV	*t*(59) = 25.79	<0.001	*
Blink Absent	FPz	ICA–RLS-AF	−2.912 μV	0.758 μV	*t*(59) = −29.737	<0.001	*
		ICA–AFFiNE	−0.265 μV	0.690 μV	*t*(59) = −2.977	0.004	*
		RLS-AF–AFFiNE	2.646 μV	0.872 μV	*t*(59) = 23.504	<0.001	*
	Pz	ICA–RLS-AF	−0.889 μV	0.378 μV	*t*(59) = −18.209	<0.001	*
		ICA–AFFiNE	−0.131 μV	0.292 μV	*t*(59) = −3.469	0.001	*
		RLS-AF–AFFiNE	0.758 μV	0.192 μV	*t*(59) = 30.582	<0.001	*

**Table 2 bioengineering-11-00707-t002:** Paired *t*-test results for absolute errors in the oddball and difference wave amplitude measures resulting from each Correction Method in the Blink Absent trials. Latency measures were excluded from this test as their distribution was bimodal; practically speaking, latency error was either zero or one sample point (i.e., 3.9 ms), with rare exceptions, across all Correction Methods. Asterisks indicate differences significant at the 0.05 level.

Measure	Waveform	Test Pairs	M	SD	Statistic	*p*	
Amplitude	Oddball	ICA–RLS-AF	−0.029 μV	0.141 μV	*t*(16) = −0.838	0.414	
		ICA–AFFiNE	−0.012 μV	0.149 μV	*t*(16) = −0.345	0.735	
		RLS-AF–AFFiNE	0.016 μV	0.023 μV	*t*(16) = 2.567	0.021	*
	Difference	ICA–RLS-AF	−0.022 μV	0.244 μV	*t*(16) = −0.38	0.709	
		ICA–AFFiNE	0.002 μV	0.243 μV	*t*(16) = 0.035	0.973	
		RLS-AF–AFFiNE	0.025 μV	0.043 μV	*t*(16) = 2.361	0.031	*

**Table 3 bioengineering-11-00707-t003:** Paired *t*-test results for absolute errors in the oddball and difference wave amplitude and latency measures resulting from each Correction Method in the Blink Present trials.

Measure	Waveform	Test Pairs	M	SD	Statistic	*p*
Amplitude	Oddball	ICA–RLS-AF	0.178 μV	0.941 μV	*t*(16) = 0.78	0.447
		ICA–AFFiNE	0.368 μV	1.212 μV	*t*(16) = 1.253	0.228
		RLS-AF–AFFiNE	0.190 μV	0.615 μV	*t*(16) = 1.276	0.220
	Difference	ICA–RLS-AF	−0.079 μV	0.374 μV	*t*(16) = −0.873	0.395
		ICA–AFFiNE	−0.105 μV	0.404 μV	*t*(16) = −1.075	0.298
		RLS-AF–AFFiNE	−0.026 μV	0.083 μV	*t*(16) = −1.299	0.212
Latency	Oddball	ICA–RLS-AF	−1.379 ms	16.684 ms	*t*(16) = −0.341	0.738
		ICA–AFFiNE	1.149 ms	17.727 ms	*t*(16) = 0.267	0.793
		RLS-AF–AFFiNE	2.528 ms	8.164 ms	*t*(16) = 1.277	0.220
	Difference	ICA–RLS-AF	−1.608 ms	7.820 ms	*t*(16) = −0.848	0.409
		ICA–AFFiNE	−1.838 ms	8.527 ms	*t*(16) = −0.889	0.387
		RLS-AF–AFFiNE	−0.230 ms	4.020 ms	*t*(16) = −0.236	0.817

**Table 4 bioengineering-11-00707-t004:** Real data oddball P300 amplitudes [μV] of individual participants.

Artifact Condition	Participant	Correction Method			
		Unfiltered	ICA	RLS-AF	AFFiNE
Blink Present	1	9.55	6.12	5.72	6.17
	2	9.70	5.63	5.57	5.69
	3	10.09	6.50	5.40	5.99
	4	11.73	7.94	8.19	8.48
	5	12.19	8.09	6.43	6.68
	6	12.63	5.88	5.86	5.88
	7	21.31	15.24	16.28	16.35
	8	24.86	10.75	9.74	9.88
	9	9.00	5.19	3.98	3.97
	10	19.15	10.46	9.20	9.25
	11	12.85	9.48	8.54	8.51
	12	13.78	10.47	10.12	10.13
	13	22.36	19.17	12.83	15.87
	14	12.89	3.68	5.39	6.00
	15	8.76	3.61	3.85	3.88
	16	16.84	11.80	11.29	11.54
	17	13.62	5.27	6.96	7.32
Blink Absent	1	8.03	8.24	8.08	8.08
	2	9.54	9.52	9.22	9.20
	3	7.20	7.27	7.12	7.10
	4	10.84	10.68	10.71	10.75
	5	6.21	6.54	6.40	6.36
	6	4.95	5.21	5.35	5.34
	7	7.31	7.56	7.26	7.29
	8	4.58	4.69	4.54	4.55
	9	0.47	0.60	0.63	0.64
	10	11.46	11.62	11.02	11.04
	11	8.84	8.76	8.75	8.76
	12	7.69	7.69	7.51	7.51
	13	15.51	15.56	15.32	15.41
	14	7.26	7.62	7.53	7.49
	15	3.44	3.47	3.39	3.41
	16	10.76	10.77	10.83	10.82
	17	5.65	5.60	5.57	5.57

**Table 5 bioengineering-11-00707-t005:** Real data oddball P300 latencies [ms] of individual participants.

Artifact Condition	Participant	Correction Method			
		Unfiltered	ICA	RLS-AF	AFFiNE
Blink Present	1	625.00	492.19	488.28	500.00
	2	511.72	421.88	421.88	425.78
	3	562.50	402.34	375.00	394.53
	4	511.72	472.66	492.19	496.09
	5	464.84	460.94	457.03	457.03
	6	644.53	445.31	441.41	441.41
	7	500.00	480.47	492.19	488.28
	8	574.22	484.38	460.94	460.94
	9	523.44	476.56	468.75	464.84
	10	527.34	468.75	460.94	460.94
	11	550.78	523.44	519.53	519.53
	12	484.38	437.50	433.59	433.59
	13	476.56	457.03	410.16	433.59
	14	523.44	402.34	429.69	433.59
	15	476.56	355.47	363.28	363.28
	16	539.06	445.31	437.50	441.41
	17	601.56	453.13	496.09	500.00
Blink Absent	1	484.38	484.38	484.38	484.38
	2	484.38	484.38	484.38	480.47
	3	464.84	460.94	457.03	460.94
	4	476.56	472.66	476.56	476.56
	5	417.97	421.88	421.88	421.88
	6	445.31	449.22	453.13	453.13
	7	453.13	453.13	449.22	449.22
	8	394.53	398.44	394.53	394.53
	9	425.78	425.78	421.88	429.69
	10	449.22	449.22	449.22	449.22
	11	472.66	472.66	472.66	472.66
	12	445.31	441.41	441.41	441.41
	13	433.59	429.69	429.69	429.69
	14	453.13	457.03	457.03	457.03
	15	367.19	367.19	367.19	367.19
	16	449.22	449.22	453.13	449.22
	17	449.22	445.31	445.31	449.22

**Table 6 bioengineering-11-00707-t006:** Real data difference wave P300 amplitudes [μV] of individual participants.

Artifact Condition	Participant	Correction Method			
		Unfiltered	ICA	RLS-AF	AFFiNE
Blink Present	1	8.73	9.54	9.92	10.11
	2	4.46	4.29	4.59	4.62
	3	5.99	6.02	5.91	6.00
	4	9.82	10.02	9.65	9.64
	5	9.56	10.12	9.52	9.63
	6	3.59	4.87	4.93	4.90
	7	12.50	12.76	12.57	12.51
	8	10.36	9.77	9.80	9.81
	9	2.38	2.20	2.10	2.13
	10	7.99	7.82	8.13	8.05
	11	10.67	11.45	12.29	12.35
	12	7.39	7.98	8.56	8.61
	13	11.35	11.62	11.17	11.04
	14	5.07	4.52	4.64	4.57
	15	0.53	0.81	0.64	0.67
	16	9.54	10.02	10.26	10.42
	17	4.49	5.08	4.52	4.49
Blink Absent	1	7.54	7.93	8.10	8.09
	2	9.29	9.28	8.96	8.96
	3	7.09	7.01	6.86	6.86
	4	11.55	11.40	11.53	11.55
	5	5.71	6.25	6.14	6.08
	6	5.21	5.06	5.46	5.46
	7	5.14	5.02	5.04	5.10
	8	3.22	3.39	3.26	3.27
	9	1.22	1.28	1.65	1.64
	10	8.51	9.29	8.59	8.56
	11	7.43	7.33	7.34	7.32
	12	6.57	6.56	6.38	6.39
	13	13.26	13.17	12.89	13.05
	14	9.58	9.97	9.96	9.91
	15	1.65	1.69	1.59	1.63
	16	10.81	10.70	10.72	10.71
	17	4.97	5.18	5.08	5.07

**Table 7 bioengineering-11-00707-t007:** Real data difference wave P300 latencies [ms] of individual participants.

Artifact Condition	Participant	Correction Method			
		Unfiltered	ICA	RLS-AF	AFFiNE
Blink Present	1	570.31	519.53	523.44	531.25
	2	464.84	410.16	417.97	421.88
	3	382.81	382.81	382.81	386.72
	4	570.31	531.25	539.06	542.97
	5	500.00	507.81	503.91	503.91
	6	449.22	460.94	460.94	460.94
	7	492.19	503.91	507.81	507.81
	8	472.66	437.50	433.59	437.50
	9	675.78	492.19	480.47	484.38
	10	621.09	488.28	480.47	484.38
	11	515.63	523.44	539.06	542.97
	12	464.84	449.22	445.31	445.31
	13	441.41	433.59	414.06	421.88
	14	406.25	402.34	398.44	394.53
	15	351.56	324.22	320.31	320.31
	16	421.88	410.16	414.06	417.97
	17	460.94	445.31	449.22	449.22
Blink Absent	1	488.28	488.28	488.28	488.28
	2	488.28	488.28	484.38	484.38
	3	464.84	460.94	460.94	460.94
	4	515.63	515.63	515.63	515.63
	5	433.59	437.50	437.50	437.50
	6	480.47	480.47	484.38	484.38
	7	449.22	449.22	445.31	445.31
	8	371.09	375.00	371.09	371.09
	9	460.94	460.94	460.94	464.84
	10	484.38	488.28	488.28	488.28
	11	468.75	468.75	468.75	468.75
	12	484.38	484.38	480.47	480.47
	13	453.13	449.22	449.22	453.13
	14	445.31	449.22	449.22	449.22
	15	363.28	363.28	363.28	363.28
	16	453.13	453.13	453.13	453.13
	17	453.13	453.13	453.13	453.13

## Data Availability

The datasets presented in this article are not readily available because they have not been approved for public release by the United States Air Force. Requests to access the datasets should be directed to Justin R. Estepp.

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
