# Peer review of "Adaptive Filtering with Fitted Noise Estimate (AFFiNE): Blink Artifact Correction in Simulated and Real P300 Data"

_bioengineering, 2024, doi:10.3390/bioengineering11070707_

Round 1

Reviewer 1 Report

Comments and Suggestions for Authors

Dear Authors, the manuscript Adaptive Filtering with Fitted Noise Estimate (AFFiNE): Blink Artifact Correction in Simulated and Real P300 Data, Manuscript ID: bioengineering-3038505, has some weaknesses that must be revised in a required manner.

Please refer to the below comments:

1.      The length of the Abstract section is too long. The Authors must put most of the significant information in this part.

2.      In the same section, the main advantages and proposals of the study must be highlighted.

3.      The Introduction section must be significantly improved. Firstly, the motivation is hidden or unclear. Secondly, must be derived from the lack in the current state of knowledge, which is usually received by a critical review of the literature. Since, in the current form, the critical review of the literature is negligible, the motivation from lines 166-170 is not received from it and the reader can be lost.

4.      In one of the parts of subsection 2.1, the main flow chart of the experiment must be provided so that it is difficult to restrict of what is the main line of the study.

5.      It does not justify why the preprocessing of the data is applied. The subsection 2.3.1. Pre-processing is too weak and not suitably motivated.

6.      What are the main advantages and disadvantages of the method presented in the subsection: 2.3.2. Artifact Correction Methods? Any limitations? Further, there is no critical discussion.

7.      The meaning of the P300 Measurements must be emphasized, e.g. in subsection 2.3.5. The Authors do not appropriately indicate the main advantage of the method.

8.      The comparison of all three Correction Methods must be discussed more comprehensively in section 3.2.4.

9.      Some more detailed justifications should be addressed in the 4.4. Differences Between Simulated and Real Data section or, respectively, Authors should add The Outlook.

10.  The Conclusions should be divided into separate and numbered gaps. Moreover, the rough results must be out of the main proposals. Finally, the main proposal must be highlighted and the Authors must show the reader the significance of the final, main conclusion.

From the above, the reviewed manuscript must be improved significantly before any further processing of the BioEngineering journal, if allowed by the Editor.

Author Response

We would like to thank the reviewer for their reconsideration of the revised manuscript. The reviewer’s concerns have been all addressed, and revisions have been marked using track changes in the word file.

  1. The length of the Abstract section is too long. The Authors must put most of the significant information in this part.

Authors’ Response: We have reduced the abstract and confirmed it is within the required 200-word count required by the journal. The abstract now only contains the most significant points of the paper, leaving the details to the manuscript body.

  1. In the same section, the main advantages and proposals of the study must be highlighted.

Authors’ Response: The main advantages of AFFiNE are now summarized in the Conclusions section of the abstract: “AFFiNE is an ocular artifact correction technique that is implementable in on-line analyses; it can adapt to non-stationarity and is independent of channel density and recording duration. AFFiNE can be utilized for the removal of blink artifact in situations where ICA may not be practically or theoretically useful.”

  1. The Introduction section must be significantly improved. Firstly, the motivation is hidden or unclear. Secondly, must be derived from the lack in the current state of knowledge, which is usually received by a critical review of the literature. Since, in the current form, the critical review of the literature is negligible, the motivation from lines 166-170 is not received from it and the reader can be lost.

Authors’ Response: We now recognize that Lines 166-170 were not best placed at the end of the Introduction. We have now moved them to the Materials and Methods section of the manuscript. This facilitated the motivation of the study to be summarized in the final paragraph of the Introduction, as would typically be expected. We also expanded our literature review to include prior art that was not previously discussed.

  1. In one of the parts of subsection 2.1, the main flow chart of the experiment must be provided so that it is difficult to restrict of what is the main line of the study.

Authors’ Response: We have added a flowchart of the study, now in Figure 2.

  1. It does not justify why the preprocessing of the data is applied. The subsection 2.3.1. Preprocessing is too weak and not suitably motivated.

Authors’ Response: We have added a citation to justify the use of the preprocessing steps applied in this section. “This filter range removes low frequency drifts and high frequency artifacts while retaining neural information and preserving the ERP shape (Tanner, Morgan-Short, & Luck, 2015).”

  1. What are the main advantages and disadvantages of the method presented in the subsection:

2.3.2. Artifact Correction Methods? Any limitations? Further, there is no critical discussion.

Authors’ Response: The Introduction section (in the original manuscript, Lines 57-139) contains a critical discussion of the advantages, disadvantages, and limitations of each method, which are the motivation for developing the AFFiNE algorithm.

  1. The meaning of the P300 Measurements must be emphasized, e.g. in subsection 2.3.5. The Authors do not appropriately indicate the main advantage of the method.

Authors’ Response: We have added a clarifying statement to indicate the advantage of these methods (and made small edits to the ordering of the original text to accommodate): “[these measures] were selected because they reliably quantify the amplitude and latency of the P300 component without being sensitive to high-frequency noise and overlapping negative components.”

  1. The comparison of all three Correction Methods must be discussed more comprehensively in section 3.2.4.

Authors’ Response: Because there were no statistically significant differences in latency and amplitude measurements between Methods in the Blink Present condition, the results in section 3.2.4 were very straightforward to discuss, which is likely why this section is shorter and may seem less comprehensive than the other subsections in our results. As stated in the closing sentence of section 3.2.4, ‘Correction Methods performed equally (for both amplitude and latency) at removing blink artifact in the Blink Present condition.’ We had previously included a discussion of these results in subsection 4.2 in our Discussion.

  1. Some more detailed justifications should be addressed in the 4.4. Differences Between Simulated and Real Data section or, respectively, Authors should add The Outlook.

Authors’ Response: From this comment, we recognize that we did not sufficiently state a definitive discussion point for sections 4.3 and 4.4. We have added this at the end of 4.4. to summarize our intent of these two sections: “Although the design of our simulated dataset was well-motivated for this purpose, from the simulated data alone, it would be possible to come to the incorrect conclusion that blink artifacts are sufficiently mitigated in difference-wave analyses; we highlight this point to further illustrate that, despite their benefits, simulated data often have unintentional limitations that can be elucidated when compared to real data. Where possible, the use of both simulated and real data in the development and evaluation of signal processing techniques is a recommended best practice.”

  1. The Conclusions should be divided into separate and numbered gaps. Moreover, the rough results must be out of the main proposals. Finally, the main proposal must be highlighted and the Authors must show the reader the significance of the final, main conclusion.

Authors’ Response: In lieu of numbering the conclusions, we have referenced them to sections of the manuscript where they are primarily discussed and shown in results; in this way, the reader can very easily follow both the main proposals and conclusions by referring back to the most important sections of the Results and Discussion. The final sentence in the manuscript is the final, main conclusion: “ In sum, AFFiNE is a practical alternative to ICA for online data analysis and a theoretical alternative to ICA in situations where data stationarity may be a concern.”

Additional Notes from the Authors:

  1. We noticed an incorrectly numbered/labeled heading in the original submission on Line 402; ‘2.3. Data Analysis’ should have been ‘2.4. Statistical Analysis.’ It has been corrected.
  2. We noticed an incorrectly numbered/labeled heading in the original submission on Line 610; ‘3.2.4. Measurement Errors in the Blink Absent Condition’ should have been ‘3.2.4. Measurement Errors in the Blink Present Condition.’ It has been corrected.
  3. There was an errant section heading on Line 632 in the original manuscript; it has been deleted.
  4. Minor changes to text to improve readability were made throughout.
  5. Changes to figure and reference numbers that resulted from revisions have already been made.

Reviewer 2 Report

Comments and Suggestions for Authors

This manuscript validates a new artifact correction technique (AFFiNE) for EEGs. Some issues might be clarified, in particular with respect to any assumptions in designing/tuning the adaptive filters:

1. AFFiNE is stated to fit splines to the adaptive filter's reference noise input. It might be clarified as to whether this reference noise input is also available to/used by the existing ICA method.

2. In Line 123, it is stated that the improvement offered by BARS over the original RLS-AF, is stated to be because "BARS allows for the spline knots to be adjusted in quantity and location". This appears to imply parameters in the implementation of BARS [filters]. If so, how were these parameters determined/implemented in BARS?

3. In Line 183 onwards, it might be clarified whether the division of participants into the real and simulated datasets was randomly done, and also any justification for the relative sizes of these two datasets, if any.

4. In Line 192 on, the standard/oddball paradigm might be briefly explained for the general audience.

5. In Section 2.2, it is explained that the total signal is the sum of three EEG signals (noise, task-relevant, artifact-relevant). It might be clarified as to how the total signal was designed to reflect actual empirical EEG signals.

Author Response

We would like to thank the reviewer for their reconsideration of the revised manuscript. The reviewer’s concerns have been all addressed, and revisions have been marked using track changes in the word file.

  1. AFFiNE is stated to fit splines to the adaptive filter's reference noise input. It might be clarified as to whether this reference noise input is also available to/used by the existing ICA method.

Authors’ Response: The individual channels used to calculate the bipolar reference noise input were included in the ICA channel space definition; we have added this clarification to section 2.3.2.

  1. In Line 123, it is stated that the improvement offered by BARS over the original RLS-AF, is stated to be because "BARS allows for the spline knots to be adjusted in quantity and location". This appears to imply parameters in the implementation of BARS [filters]. If so, how were these parameters determined/implemented in BARS?

Authors’ Response: We clarified the language in the introduction to indicate that BARS automatically adjusts spline knot quantity and location as part of the algorithm itself.

  1. In Line 183 onwards, it might be clarified whether the division of participants into the real and simulated datasets was randomly done, and also any justification for the relative sizes of these two datasets, if any.

Authors’ Response: We added clarifying language in section 2.1.1 to indicate that the participants included in the real dataset were chosen, specifically, because they contained a similar number of blink-contaminated and blink-free trials balanced across stimulation conditions (so that, later, we could equate the number of samples analyzed from each participant in the statistical analysis, which we showed in the original Figure 5). Recruitment of participants into the real dataset was done to maximize its sample size; we have also added this information to 2.1.1.

  1. In Line 192 on, the standard/oddball paradigm might be briefly explained for the general audience.

Authors’ Response: This is an excellent suggestion; we have added a brief explanation of the oddball paradigm to 2.1.2.

  1. In Section 2.2, it is explained that the total signal is the sum of three EEG signals (noise, task relevant, artifact-relevant). It might be clarified as to how the total signal was designed to reflect actual empirical EEG signals.

Authors’ Response: We added clarifying language to this effect to section 2.2.

Additional Notes from the Authors:

  1. We noticed an incorrectly numbered/labeled heading in the original submission on Line 402; ‘2.3. Data Analysis’ should have been ‘2.4. Statistical Analysis.’ It has been corrected.
  2. We noticed an incorrectly numbered/labeled heading in the original submission on Line 610; ‘3.2.4. Measurement Errors in the Blink Absent Condition’ should have been ‘3.2.4. Measurement Errors in the Blink Present Condition.’ It has been corrected.
  3. There was an errant section heading on Line 632 in the original manuscript; it has been deleted.
  4. Minor changes to text to improve readability were made throughout.
  5. Changes to figure and reference numbers that resulted from revisions have already been made.

Round 2

Reviewer 1 Report

Comments and Suggestions for Authors

Dear Authors,

All of the raised comments were respond and the manuscript improved according to the marked requirements so the manuscript in its current, revised form can be considered for publication in the Bioengineering journal.

Best regards,

Reviewer